# Training Wasserstein GANs without Gradient Penalties via Duality

## Abstract

We propose a new method for training Wasserstein generative adversarial networks (WGANs) without using gradient penalties. We develop a novel approach to accurately estimate the Wasserstein distance between datasets, specifically tailored for the generative setting. Instead of employing the commonly used gradient penalty term in the WGAN training procedure, we introduce two objective functions that utilize the $c$-transform based on Kantorovich duality, which is a fundamental property in optimal transport theory. Through our experiments, we observe that this algorithm effectively enforces the Lipschitz constraint on the discriminator, paving the way for understanding the optimal transport problem via a deep learning approach. As a result, our method provides an accurate estimation not only for the optimal discriminator but also for the Wasserstein distance between the true and generated distribution. Notably, our method eliminates the need for gradient penalties and corresponding hyperparameter tuning. Moreover, we demonstrate its effectiveness by generating competitive synthetic images using various datasets such as MNIST, Fashion-MNIST, CIFAR-10, and CelebA-HQ.

## 1 Introduction

Generative Adversarial Networks (GANs) (Goodfellow et al., 2014) have achieved remarkable success in generating synthetic images. GANs consist of two networks, the generator and the discriminator, which compete with each other. The goal of this procedure is to find a minimax solution known as Nash equilibrium. GANs have found numerous applications in machine learning, including semisupervised learning, text-to-image generation, image-to-text generation, and speech synthesis (Ledig et al., 2017; Isola et al., 2017; Zhang et al., 2017; Reed et al., 2016; Don, 2019). However, training GANs remains challenging due to the intrinsic instability associated with mode collapse, and gradient vanishing (Gulrajani et al., 2017). Several works have aimed to improve stability (Tolstikhin et al., 2017; Salimans et al., 2016; Nowozin et al., 2016; Radford et al., 2015; Kodali et al., 2017; Liu et al., 2017) and have made substantial improvements beyond the original GAN. Of particular interest in this context is the Wasserstein GAN (WGAN) framework proposed in Arjovsky et al. (2017), where the training objective for the generator network is the Wasserstein distance to the target distribution.

In terms of learning performance, experimental evidence has demonstrated the potential advantages of WGANs over GANs, eliminating the possibility of gradient vanishing and mode collapse (Arjovsky et al., 2017), but their mathematical properties have not yet been thoroughly understood. In WGANs, one aims to solve a minimax problem involving the generator and the discriminator, where the discriminator is constrained to be a 1-Lipschitz function. The original WGAN introduced weight clipping to enforce the 1-Lipschitz condition, but this approach can overly restrict the class of functions. To address this issue, WGANs with gradient penalty (WGAN-GP) were proposed in Gulrajani et al. (2017). While successful in generative settings, recent work (Milne & Nachman, 2022) shows that WGAN-GP is not trained in a way that minimizes the Wasserstein distance. It is demonstrated that the optimal value of the objective function used in WGAN-GP corresponds to the minimum cost of a congested transport model, diverging from its original philosophy (Milne & Nachman, 2022). Furthermore, several attempts have been made to under-

stand the training dynamics of WGANs by analyzing the sample complexity for estimating the Wasserstein distance as found in Mallasto et al. (2019); Stanczuk et al. (2021).

Considering the nature of WGAN, its performance relies on the proper estimation of the Wasserstein distance between two training datasets. Estimating the Wasserstein distance is of independent interest and has been extensively studied in the literature, focusing on sample complexity and efficient computation methods (Bassetti et al., 2006; Dudley, 1969; Weed & Bach, 2019). However, there is relatively limited literature on deep learning-based frameworks for estimating distance. In this paper, we investigate an efficient approach to approximating the Wasserstein distance via neural networks.

We propose an intuitive and straightforward method to learn the Wasserstein distance within the generative model framework. Our approach solely focuses on the formulation of the Wasserstein distance and does not introduce any form of penalization, eliminating the need for parameter tuning. A key feature of our proposed method is that we use sample batches to approximate the Wasserstein distance between two large, high-dimensional datasets.

Moreover, we explore avenues to enhance the stability, accuracy, and computational efficiency of WGAN training, aligning them more closely with the mathematical theory of optimal transport. Building upon Mallasto et al. (2019), we investigate the *c*-transform methods, which have shown promise in accurately estimating the true Wasserstein metric. However, these methods suffer from instability and perform suboptimally in the generative setting, as discussed in our theoretical investigation in Section 5 and Mallasto et al. (2019).

## 1.1 Related work

### 1.1.1 Estimation of Wasserstein distance

Computing the Wasserstein distance efficiently is a long-standing problem that has attracted significant attention in both mathematics and machine learning communities. The Wasserstein distance is highly effective in quantifying the dissimilarity between two probability distributions. Cuturi (2013) introduces the concept of the Sinkhorn distance, which utilizes entropic regularization to approximate the Wasserstein distance. This approach has demonstrated remarkable computational efficiency compared to other methods such as Pele & Werman (2009). In a subsequent study by Genevay et al. (2018), the Sinkhorn distance is shown to be effective in the context of generative models. While these works address the approximate optimal transport problem and the Wasserstein distance, our paper focuses on exact optimal transport and its application to generative models.

Another closely related work is presented in Jacobs & Léger (2020), where the authors propose a novel method called the *back-and-forth method* for computing the Wasserstein distance between distributions. This method leverages the duality structure inherent in the optimal transport formulation. Not only is their algorithm efficient in computing the Wasserstein distance, but it also facilitates the determination of optimal transport maps. In our research, we exploit the structural properties of the back-and-forth method to devise a deep learning algorithm for estimating the Wasserstein distance.

We also highlight the recent work by Xie et al. (2020), which introduces the Iterative Proximal Optimal Transport (IPOT) algorithm. The algorithm efficiently approximates the Wasserstein distance without introducing a gradient penalty term. The authors further demonstrate that IPOT can also be applied to generative tasks. However, unlike our method, which directly enforces the 1-Lipschitz condition using Kantorovich duality and the *c*-transform, IPOT focuses on optimizing the transport plan without explicitly addressing the Lipschitz continuity of the discriminator.

### 1.1.2 Enforcement of 1-Lipshitz continuity

At the core of the implementation of Kantorovich-Rubinstein duality for estimating Wasserstein distance and the framework of WGAN training is the enforcement of 1-Lipschitz continuity for an optimal discriminator $\phi$, which requires ensuring that the gradient norm of $\phi$ is 1 almost everywhere, i.e., $\|D\phi\| = 1$ where $\|\cdot\|$ denotes the standard Euclidean norm throughout the paper. Traditionally, this involves explicitly computing

the discriminator's gradient at sample points and interpolated sample points. However, Wei et al. (2018) argues that applying the gradient penalty only at sample points is insufficient. Additionally, Petzka et al. (2017) emphasizes that globally satisfying $\|D\phi\| = 1$ for an optimal discriminator can hinder optimization. To address these challenges, they propose penalizing only gradient norms above 1 within a manifold around the sample points, offering a stable approximation of the Wasserstein distance.

Another notable approach is introduced in Miyato et al. (2018), where the authors propose spectral normalization to enforce Lipschitz continuity for functions parameterized by deep neural networks.

In contrast, our approach does not include any penalization term that enforces the condition $\|D\phi\| = 1$ on random samples obtained by interpolating real and synthetic data points. Therefore, our framework eliminates the need for a weight parameter $\lambda$ that controls the intensity of the penalization.

### 1.2 Our contribution

We propose a novel and stable training method for Wasserstein GANs (WGANs) that is inspired by the back-and-forth method for the optimal transport problem on spatial grids, as introduced by Jacobs & Léger (2020). In traditional WGANs, the objective function for training the discriminator and the generator is defined using the Kantorovich duality formulation of the 1-Wasserstein distance between two probability measures $\mu$ and $\nu$ on $\Omega \subset \mathbb{R}^d$, which is given by

$$W_1(\mu, \nu) = \sup_{\phi \in \text{Lip}(\Omega)} \left( \mathbb{E}_\mu[\phi] - \mathbb{E}_\nu[\phi] \right). \tag{1.1}$$

Throughout the paper, $\text{Lip}(\Omega)$ denotes the set of 1-Lipschitz continuous functions on $\Omega$, that is, functions satisfying $|\phi(x) - \phi(y)| \le \|x - y\|$ for any $x, y \in \Omega$. Instead, we consider the following two optimization

$$\sup_{\phi \in C(\Omega)} \left( \mathbb{E}_\mu[\phi] + \mathbb{E}_\nu[\phi^c] \right) \text{ and } \sup_{\phi \in C(\Omega)} \left( \mathbb{E}_\mu[(-\phi)^c] + \mathbb{E}_\nu[-\phi] \right),$$

where $C(\Omega)$ denotes the set of continuous functions in $\Omega$ and $\phi^c$ is the $c$-transform of $\phi$ that will be discussed in Section 2.1. However, computing the $c$-transform can be computationally expensive for large high-dimensional datasets, making it challenging to use the dual forms of Wasserstein distance. Our method combines both formulations to develop a deep learning framework for estimating the 1-Wasserstein distance between large high-dimensional datasets, where the standard back-and-forth method is unavailable. Additionally, our algorithm optimizes the traditional dual forms of Wasserstein distance.

One of the key advantages of our method is that it achieves stability in WGAN training without explicitly penalizing the discriminator gradients, which is often required in approaches motivated by WGAN-GP. This eliminates the need for hyperparameter tuning associated with gradient penalties whose mathematical justification is unknown to the best of our knowledge.

Our contributions can be summarized as follows:

- (Section 3 & Algorithm 1) Based on the theoretical analysis, we propose a refined approach to estimating the Wasserstein distance between large high-dimensional datasets using the $c$-transform. We empirically verify the 1-Lipschitz condition without the need for a gradient penalty term, demonstrating its effectiveness through various examples.

- (Section 4 & Algorithm 2) Leveraging the accurate estimation of the Wasserstein distance, we design a stable WGAN training algorithm that enforces the correct 1-Lipschitz bound without the need for a gradient penalty. This eliminates the requirement for hyperparameter tuning associated with the penalty term.

- (Section 5) Our algorithm produces high-quality synthetic images and performs well with various types of data, including mixtures of Gaussians, MNIST, Fashion-MNIST, CIFAR-10, and high-dimensional datasets such as CelebA-HQ. The algorithm accurately computes the discriminator $\phi$ during training and the 1-Wasserstein distance between the target and the distribution generated.

## 2 Preliminaries

### 2.1 Optimal transport overview

For the $d$-dimensional Euclidean space $\mathbb{R}^d$, let $\mathcal{P}(\Omega)$ be the space of Borel probability measures on $\Omega \in \mathbb{R}^d$. The $p$-Wasserstein distance between two probability measures $\mu$ and $\nu$ in $\mathcal{P}(\Omega)$ is defined as

$$W_p(\mu, \nu) := \left\{ \inf_{\gamma \in \Pi(\mu, \nu)} \int_{\Omega \times \Omega} \|x - y\|^p \mathrm{d}\gamma \right\}^{\frac{1}{p}}. \tag{2.1}$$

Here, $\Pi(\mu, \nu)$ is the set of all Borel probability measures $\pi$ on $\Omega \times \Omega$ such that $\pi(A \times \Omega) = \mu(A)$ and $\pi(\Omega \times A) = \nu(A)$ for all measurable subsets $A \subset \Omega$.

Under suitable assumptions on $\mu$ and $\nu$, we also have

$$W_p(\mu, \nu) = \left\{ \inf_{T \# \mu = \nu} \int_{\Omega} \|x - T(x)\|^p \mathrm{d}\mu(x) \right\}^{\frac{1}{p}} \tag{2.2}$$

where a pushforward measure is defined as $T \# \rho(A) := \rho(T^{-1}(A))$ for a set $A \subset \Omega$ and $\rho \in \mathcal{P}(\Omega)$. It can be interpreted as the minimum cost of transporting the distribution $\mu$ to $\nu$ with respect to the metric given by $\|x - y\|^p$ for $p \geq 1$, which also can be used as the *distance* between two distributions. However, solving this optimization problem is challenging in general.

Here, we give a formal derivation of the Kantorovich-Rubenstein duality. Introducing the Lagrange multiplier $\phi : \Omega \to \mathbb{R}$ which enforces the condition $T \# \mu = \nu$, and interchanging inf and sup, (2.2) can be reformulated as

$$\begin{aligned} W_p^p(\mu, \nu) &= \inf_T \sup_\phi \left\{ \int_\Omega \|x - T(x)\|^p \, d\mu(x) + \int_\Omega \phi \, d\nu - \int_\Omega (\phi \circ T) \, d\mu \right\} \\ &= \sup_\phi \inf_T \left\{ \int_\Omega \|x - T(x)\|^p \, d\mu(x) + \int_\Omega \phi \, d\nu - \int_\Omega (\phi \circ T) \, d\mu \right\} \\ &= \sup_\phi \left\{ \int_\Omega \phi \, d\nu + \int_\Omega \inf_T \left\{ \|x - T(x)\|^p - (\phi \circ T) \right\} d\mu(x) \right\}. \end{aligned}$$

Motivated by this, let us introduce the $c$ transform, $\phi^c(y) := \inf_{x \in \Omega} \left\{ \|x - y\|^p - \phi(x) \right\}$ for $y \in \Omega$ (see Proposition 1.11 in Santambrogio (2015)). Finally, the Kantorovich-Rubenstein duality yields that

$$W_p^p(\mu, \nu) = \sup_{\phi \in C(\Omega)} \left\{ \int_\Omega \phi \mathrm{d}\mu + \int_\Omega \phi^c \mathrm{d}\nu \right\}^{\frac{1}{p}}. \tag{2.3}$$

Consequently, the duality property allows for a more efficient way of computing $W_p(\mu, \nu)$ as one can avoid explicitly constructing $\gamma(x, y)$. Instead of optimizing over the infinite-dimensional space of couplings $\gamma(x, y)$, we optimize over a scalar function $\phi(x)$, which is known to be efficient in high dimensions.

Additionally, one can derive another representation for this formulation when $p = 1$, as in Remark 6.5 of Villani (2008).

**Proposition 1.** *Let $\mu, \nu \in \mathcal{P}(\Omega)$. The 1-Wasserstein distance (1.1) between two probability measures $\mu$ and $\nu$ is given by*

$$W_1(\mu, \nu) = \sup_{\phi \in \mathrm{Lip}(\Omega)} \left\{ \int_\Omega \phi \mathrm{d}(\mu - \nu) \right\}. \tag{2.4}$$

*Proof.* See Villani (2008). □

The formulation (2.4) is useful not only for the development of an algorithm estimating the Wasserstein distance between distributions but also for the generative model. In the framework of WGANs, $\phi$ plays the role of discriminators, and $\nu$ is the generative distribution. The objective of WGAN is to find $\nu$ from a parametrized set of pushforward maps of a noise source that best mimics $\mu$, which is given. In practice, the integral is replaced by a sample average, and optimization is conducted iteratively using mini-batches of samples.

## 2.2 The objective functions of Wasserstein GANs and its variants

For $0 < m \ll d$ and $p \geq 1$, let $\mu \in \mathcal{P}(\Omega)$ denote the distribution of data, $\Omega \subset \mathbb{R}^d$ and $\rho \in \mathcal{P}(\mathbb{R}^m)$ be a source distribution such as Gaussian noise. The main goal of the Wasserstein GANs is to find a parametrized generator $G_\theta : \mathbb{R}^m \to \mathbb{R}^d$, that minimizes $W_p(\mu, G_\theta \# \rho)$.

Here, the pushforward measure $G_\theta \# \rho$ is defined such that $(G_\theta \# \rho)(B) = \rho(G_\theta^{-1}(B))$ for all measurable sets $B \subset \mathbb{R}^d$. Therefore, we solve the following minimax problem:

$$\inf_\theta \left\{ \sup_\eta \left\{ \int_\Omega \phi_\eta (\mathrm{d}\mu - \mathrm{d}G_\theta \# \rho) : \phi_\eta \in \mathrm{Lip}_1 \right\} \right\},$$

where the discriminator $\phi_\eta$ and the generator $G_\theta$ are parametrized by neural networks. As briefly described in the introduction, enforcing the Lipschitz condition is a challenging task in the implementation of WGANs, particularly for large high-dimensional datasets. Various regularization tricks have been introduced in the literature to resolve this issue. For instance, the loss function of WGAN-GP introduced in Gulrajani et al. (2017) is given as follows:

$$\inf_\theta \sup_\eta \left\{ \int_\Omega \phi_\eta (\mathrm{d}\mu - \mathrm{d}G_\theta \# \rho) + \lambda \int_\Omega \left( \|D\phi_\eta\| - 1 \right)^2 \mathrm{d}\omega \right\}. \qquad \text{(WGAN-GP)}$$

A central aspect of this approach is to enforce the condition $\|D\phi_\eta\| = 1$ throughout the entire domain, although this is not a necessary and sufficient condition for (2.4) to attain equality. There also have been several attempts to design a loss function trying to enforce 1-Lipschitzness such as WGAN-LP (Petzka et al., 2017), WGAN-CT (Wei et al., 2018), and spectral normalizations (Miyato et al., 2018), but all use WGAN-GP as the baseline by inserting the gradient penalty term in the loss function.

On the other hand, the $c$-transform-based method has been proposed in Mallasto et al. (2019). The objective function is given as

$$\inf_\theta \sup_\eta \left\{ \int_\Omega \phi_\eta \mathrm{d}\mu - \int_\Omega \phi_\eta^c \mathrm{d}(G_\theta \# \rho) \right\}, \qquad (c\text{-transform})$$

where $\phi_\eta^c$ is the $c$-transform of $\phi_\eta$ defined in (3.4) below. This does not have the gradient penalty term, but as investigated in Mallasto et al. (2019) and Stanczuk et al. (2021), the method does not perform well in the generative setting. More discussions can be found in Section 4.3.

## 3 Estimation of 1-Wasserstein distance

In this section, we present a theoretical framework for a new algorithm designed not only for the estimation of Wasserstein distances but also for training WGANs without the need for gradient penalties. Our approach involves a novel learning method that accurately estimates the 1-Wasserstein distance between large, high-dimensional datasets. Subsequently, the question of whether WGANs create a distribution mimicking the true distribution can be answered.

While various approaches have been previously discussed, we extend the concept of the back-and-forth method to the realm of deep learning, enabling us to effectively estimate the 1-Wasserstein distance.

Building upon this proposed method, we introduce a new training technique for WGANs in the following section. To accomplish this, we first introduce the notion of a universal admissible condition.

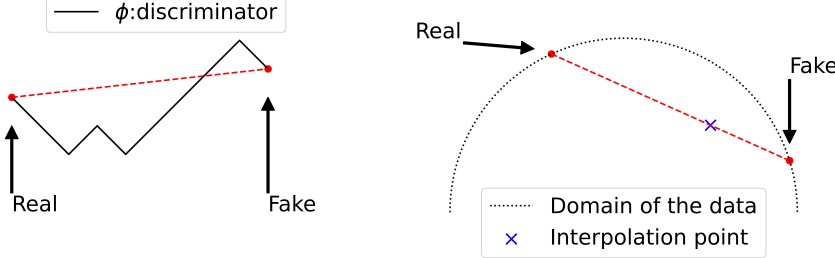

Figure 1: Why does GP fail to enforce 1-Lipschitz regularity in the discrete setting? When the probability measures are discrete, forcing $\|D\phi\| = 1$ along random interpolation points between the real and fake data may induce underestimate of the Lipschitz constant of $\phi$, $\sup_{x \sim \text{real}, y \sim \text{fake}} |\frac{\phi(x)-\phi(y)}{\|x-y\|}|$. The left shows that the Lipschitz constant between real and fake data is less than 1, as depicted by the dashed red line. In high dimensional space like the diagram on the right, WGAN-GP simply imposes the gradient norm constraint on a random interpolation point denoted by blue, which does not help impose the 1-Lipschitz continuity of the discriminator while our algorithm will directly compute the slope between these two points.

## 3.1 The universal admissible condition

To find a probability measure $\nu$ that best mimics $\mu$ using mini-batches, we first define the so-called universal admissible set as the set of $\phi$'s satisfying

$$\phi(x) - \phi(y) \le \|x - y\| \text{ for all } (x, y) \in \text{supp}(\mu) \times \text{supp}(\nu) \tag{3.1}$$

for two probability measures $\mu, \nu \in \mathcal{P}(\Omega)$.

The aforementioned condition offers two primary advantages: a reduced computational cost and an expanded admissible set. The computational efficiency of utilizing (3.1) is more prominent when the support sets of $\mu$ and $\nu$, denoted as $\text{supp}(\mu)$ and $\text{supp}(\nu)$ respectively, are relatively small. This is particularly evident in scenarios where the data points lie in a low-dimensional manifold.

Conversely, any function on $\Omega$ that satisfies the 1-Lipschitz property also satisfies the relation (3.1). Hence, our condition is less restrictive than enforcing strict 1-Lipschitzness on $\Omega$. Consequently, (3.1) allows for the representation of a broader class of functions.

To discuss the potential advantages of the utilization of the universal admissible condition, we point out the limitations of imposing the gradient penalty. Fig. 1 demonstrates why training with the gradient penalty fails to estimate the true Wasserstein distance between distributions accurately. The core of the WGAN-GP training algorithm involves selecting points a pair of real and synthetic data points, followed by selecting a random interpolation point, where the norm of the gradient is enforced to be 1. However, as depicted in Fig. 1, this approach tends to underestimate the slope, particularly in high dimensional-spare datasets, even if the norm of the gradient at discrete points is forced to be constant. An alternative approach, spectral normalization (Miyato et al., 2018), which enforces the Lipschitz constant to be a constant everywhere, possesses a similar limitation. Moreover, if the distribution of real data exhibits a specific shape, as shown on the right of Fig. 1, such a training method that relies on interpolation of two points can fail to impose 1-Lipschitz constraint of the discriminator $\phi$.

Using the duality formulation, we prove that the relaxation of the set of feasible functions still computes the Wasserstein distance. While the proof is straightforward from the duality structure, the observation gives us a clue for the construction of our new algorithm. Instead of the Lipschitz condition on $\Omega$, it suffices to enforce (3.1) while training WGANs. This observation plays a critical role in one of the main algorithms, Algorithm 1.

**Theorem 1.** *For any $\mu, \nu \in \mathcal{P}(\Omega)$, we have*

$$W_1(\mu, \nu) = \sup\left\{ \int_\Omega \phi \mathrm{d}(\mu - \nu) : \phi \in C(\Omega) \text{ satisfies (3.1)} \right\}. \tag{3.2}$$

*Proof.* Recall the duality formulation (2.4) of $W_1$. As all 1-Lipschitz functions satisfy (3.1), we have

$$W_1(\mu, \nu) \leq \sup\left\{ \int_\Omega \phi \mathrm{d}(\mu - \nu) : \phi \in C(\Omega) \text{ satisfies (3.1)} \right\}.$$

On the other hand, consider a so-called transport plan,

$$\gamma : \Omega \times \Omega \to \mathbb{R},$$

which satisfies

$$\gamma(A \times \Omega) = \mu(A) \text{ and } \gamma(\Omega \times A) = \nu(A) \tag{3.3}$$

for all measurable subsets $A \subset \Omega$. Since $\mathrm{supp}(\gamma) \subset \mathrm{supp}(\mu) \times \mathrm{supp}(\nu)$, we have

$$\int_\Omega \phi(\mathrm{d}\mu - \mathrm{d}\nu) = \int_{\Omega \times \Omega} (\phi(x) - \phi(y))\mathrm{d}\gamma(x, y),$$
$$\leq \int_{\Omega \times \Omega} \|x - y\| \mathrm{d}\gamma$$

for all $\phi$ satisfying (3.1) and any transport plan $\gamma$ satisfying (3.3).

As a consequence of this inequality and (2.1), we have

$$\sup\left\{ \int_\Omega \phi(\mathrm{d}\mu - \mathrm{d}\nu) : \phi \in C(\Omega) \text{ satisfies (3.1)} \right\} \leq \inf_{\gamma \in \Pi(\mu,\nu)} \left\{ \int_{\Omega \times \Omega} \|x - y\| \mathrm{d}\gamma(x, y) \right\}$$
$$= W_1(\mu, \nu).$$

Thus, the result (3.2) follows. □

### 3.2 Our approach: comparison between pseudo-objective functions

Given two probability measures $\mu, \nu \in \mathcal{P}(\Omega)$, we recall the dual forms, (2.3) and (2.4) to introduce following pseudo-objective functions,

$$\mathcal{J}_1(\phi; \mu, \nu) = \int_\Omega \phi \mathrm{d}\mu + \int_\Omega (-\phi)\mathrm{d}\nu,$$
$$\mathcal{J}_2(\phi; \mu, \nu) = \int_\Omega \phi \mathrm{d}\mu + \int_\Omega \phi^c(\cdot; \mu)\mathrm{d}\nu,$$
$$\mathcal{J}_3(\phi; \mu, \nu) = \int_\Omega (-\phi)^c(\cdot; \nu)\mathrm{d}\mu + \int_\Omega (-\phi)\mathrm{d}\nu,$$
$$\mathcal{J}_4(\phi; \mu, \nu) = \int_\Omega (-\phi)^c(\cdot; \nu)\mathrm{d}\mu + \int_\Omega \phi^c(\cdot; \mu)\mathrm{d}\nu,$$

where the *pseudo c-transform* of $\omega \in \mathcal{P}(\Omega)$ is defined by

$$\phi^c(y; \omega) := \inf_{x \in \mathrm{supp}(\omega)} \{\|x - y\| - \phi(x)\} \text{ for } y \in \Omega. \tag{3.4}$$

An important feature of the formulation is that we shall take the infimum in the $c$-transform over the support of the probability measure instead of the whole domain $\Omega$, which is simpler to compute compared to the original $c$-transform.

This concept has been implicitly used in the literature, e.g., in the works of Farnia & Ozdaglar (2020); Mallasto et al. (2019). However, to our knowledge, the discrepancy between $\phi^c$ and $\phi^c(\cdot; \omega)$ has not been studied in detail.

To further explore the pseudo $c$-transform, let us provide some intrinsic properties of discrete $c$-transform in comparison with the original one. It is clear that the original $c$-transform satisfies $\phi^c \leq -\phi$ over $\Omega$, but this relation does not hold for $\phi^c(\cdot; \omega)$ in general. In turn, $\phi^c$ is not necessarily equal to $-\phi$ even if $\phi$ is a 1-Lipschitz function. However, the following inequalities hold for the $\mathcal{J}_i$'s, which is a key ingredient for developing our method.

**Lemma 1.** *Given two proability measures, $\mu, \nu \in \mathcal{P}(\Omega)$, assume that $\phi$ satisfies the admissibility property (3.1). Then we have*

$$\mathcal{J}_1(\phi; \mu, \nu) \leq \mathcal{J}_2(\phi; \mu, \nu) \leq \mathcal{J}_4(\phi; \mu, \nu) \text{ and}$$
$$\mathcal{J}_1(\phi; \mu, \nu) \leq \mathcal{J}_3(\phi; \mu, \nu) \leq \mathcal{J}_4(\phi; \mu, \nu).$$

*Proof.* From (3.1) and $\text{supp}(\mu) \subset \Omega$, it holds that for all $y \in \Omega$,

$$\phi^c(y; \mu) = \inf_{x \in \text{supp}(\mu)} \{\|x - y\| - \phi(x)\}$$
$$\geq \inf_{x \in \text{supp}(\mu)} \{-\phi(y)\} = -\phi(y).$$

Therefore, we conclude $\mathcal{J}_1(\phi; \mu, \nu) \leq \mathcal{J}_2(\phi; \mu, \nu)$. The other inequalities can be shown similarly. $\square$

We now demonstrate the effectiveness of leveraging the notion of pseudo $c$-transform in a batch-based learning approach.

Consider $n$ i.i.d. observations $X_1, X_2, \ldots, X_n$ distributed according to $\mu$, and another $n$ i.i.d. observations $Y_1, Y_2, \ldots, Y_n$ distributed according to $\nu$. We denote the empirical measures based on samples $X_i$'s and $Y_i$'s by $\mu_n$ and $\nu_n$ respectively:

$$\mu_n := \frac{1}{n} \sum_{i=1}^n \delta_{X_i} \text{ and } \nu_n := \frac{1}{n} \sum_{i=1}^n \delta_{Y_i}. \tag{3.5}$$

Since they are finite measures, the pseudo $c$-transforms $\mathcal{J}_1(\phi; \mu_n, \nu_n)$, $\mathcal{J}_2(\phi; \mu_n, \nu_n)$, $\mathcal{J}_3(\phi; \mu, \nu)$ and $\mathcal{J}_4(\phi; \mu_n, \nu_n)$ are well-defined and easy to compute as the optimization is performed over the finite sets, $\text{supp}(\mu_n)$ and $\text{supp}(\nu_n)$. In addition to that, we have the comparison between them as a result of Lemma 1.

We now introduce criteria in terms of empirical measures for the condition (3.1) to be satisfied. In particular, equivalence between the inequalities and (3.1) holds as follows.

**Lemma 2.** *Let $\mu, \nu \in \mathcal{P}(\Omega)$. Given $n \in \mathbb{N}$, if we have*

$$\mathcal{J}_1(\phi; \mu_n, \nu_n) \leq \mathcal{J}_2(\phi; \mu_n, \nu_n)$$

*for all $\mu_n$ and $\nu_n$, then $\phi$ satisfies the admissibility property (3.1). Here, $\mu_n$ and $\nu_n$ are empirical measures obtained from $\mu$ and $\nu$ as given in (3.5).*

*Proof.* For any $x \in \text{supp}(\mu)$ and $y \in \text{supp}(\nu)$, if $X_i = x$ and $Y_i = y$ for all $i = 1, 2, \ldots, n$, then the empirical measures are $\mu_n = \delta_x$ and $\nu_n = \delta_y$.

From $\mathcal{J}_1(\phi; \mu_n, \nu_n) \leq \mathcal{J}_2(\phi; \mu_n, \nu_n)$, we derive that

$$\phi(x) - \phi(y) \leq \phi(x) + \phi^c(y; \delta_x).$$

As $\phi^c(y; \delta_x) = \|x - y\| - \phi(x)$, we conclude that

$$\phi(x) - \phi(y) \leq \|x - y\|.$$

Hence, (3.1) holds for any $(x, y) \in \text{supp}(\mu) \times \text{supp}(\nu)$. $\square$

**Remark 1.** *The same result still holds if $\mathcal{J}_1(\phi; \mu_n, \nu_n) \leq \mathcal{J}_2(\phi; \mu_n, \nu_n)$ is replaced by $\mathcal{J}_1(\phi; \mu_n, \nu_n) \leq \mathcal{J}_3(\phi; \mu_n, \nu_n)$.*

### 3.3 Relation between $\mathcal{J}_1$ and the Wasserstein distance

Our first goal is to design a learning algorithm that estimates the Wasserstein distance between two probability measures $\mu$ and $\nu$. We begin by providing an equivalent representation beyond the Kantorovich duality formula. To motivate, we observe first that

$$\mathbb{E}_{\substack{X_i \sim \mu, Y_i \sim \nu \\ 1 \leq i \leq n}} \left[ \frac{1}{n} \sum_{i=1}^{n} (\phi(X_i) - \phi(Y_i)) \right] = \int_{\Omega} \phi \, \mathrm{d}(\mu - \nu). \tag{3.6}$$

Recalling the equivalence between the 1-Lipschitz condition and the universal admissible condition for empirical measures given in Lemma 1 and 2, we deduce the following equivalent maximization problem,

$$\sup_{\phi \in \mathcal{B}} \left\{ \mathbb{E}_{\substack{X_i \sim \mu, Y_i \sim \nu \\ 1 \leq i \leq n}} [\mathcal{J}_1(\phi; \mu_n, \nu_n)] \right\}, \tag{3.7}$$

where

$\mathcal{B} := \{ \phi \in C(\Omega) : 0 \leq \mathcal{J}_1(\phi; \mu_n, \nu_n) \leq \mathcal{J}_2(\phi; \mu_n, \nu_n)$ and
$0 \leq \mathcal{J}_1(\phi; \mu_n, \nu_n) \leq \mathcal{J}_3(\phi; \mu_n, \nu_n)$ for all empirical measures $\mu_n$ and $\nu_n$ as given in (3.5)$\}$.

Altogether, we have the following alternative formulation for achieving the Wasserstein distance between two probability measures $\mu$ and $\nu$.

**Theorem 2.** *Let $n, d \in \mathbb{N}$ be given and $\mu, \nu \in \mathcal{P}(\Omega)$ for $\Omega \subset \mathbb{R}^d$. Then, the Wasserstein distance between $\mu$ and $\nu$ is equal to the optimal value of (3.7).*

This yields that $\mathcal{J}_1$ is indeed an unbiased estimator of the Wasserstein distance for a suitable choice of $\phi$. Therefore, if trained well to find $\phi$, the functional $\mathcal{J}_1$ would yield an accurate estimate for the Wasserstein distance between two measures, $\mu$ and $\nu$.

*Proof.* For $\phi \in C(\Omega)$ and empirical measures $\mu_n$ and $\nu_n$ defined as (3.5), we have

$$\mathcal{J}_1(\phi; \mu_n, \nu_n) = \frac{1}{n} \sum_{i=1}^{n} (\phi(X_i) - \phi(Y_i)).$$

By Theorem 1, the Wasserstein distance between $\mu$ and $\nu$ is given as

$$W_1(\mu, \nu) = \sup_{\phi \in \mathrm{Lip}(\Omega)} \mathbb{E}_{\substack{X_i \sim \mu, Y_i \sim \nu \\ 1 \leq i \leq n}} [\mathcal{J}_1(\phi; \mu_n, \nu_n)]$$
$$= \sup_{\phi \text{ satisfies (3.1)}} \mathbb{E}_{\substack{X_i \sim \mu, Y_i \sim \nu \\ 1 \leq i \leq n}} [\mathcal{J}_1(\phi; \mu_n, \nu_n)].$$

By Lemma 1 and 2, $\phi$ satisfies (3.1) if and only if $\phi \in \mathcal{B}$, and this completes the proof. $\square$

To evaluate (3.7), we estimate using random samples $X_i \sim \mu$ and $Y_i \sim \nu$. Given an optimal $\phi \in \mathcal{B}$ solving (3.7), we use the law of large numbers to transform the original optimization problem into a learning problem, which is

$$\frac{1}{n} \sum_{i=1}^{n} (\phi(X_i) - \phi(Y_i)) \to \mathbb{E}_{X \sim \mu}[\phi(X)] - \mathbb{E}_{Y \sim \nu}[\phi(Y)] = W_1(\mu, \nu) \quad \text{w.p. 1 with respect to } \mu \times \nu,$$

as $n$ grows to infinity.

The error can be quantified under suitable assumptions. For instance, as long as $\phi$ is bounded, the Hoeffding inequality (Bentkus, 2004) yields that

$$\Pr \left( \left| \frac{1}{n} \sum_{i=1}^{n} (\phi(X_i) - \phi(Y_i)) - W_1(\mu, \nu)) \right| \geq \epsilon \right) \leq 2 \exp(-Cn\epsilon^2)$$

for some constant $C > 0$.

Lastly, in Algorithm 1, we use the following objective function to tackle this problem:

$$\sup_{\substack{\phi \in C(\Omega)}} \mathbb{E}_{\substack{X_i \sim \mu, Y_i \sim \nu \\ 1 \leq i \leq n}}[\mathcal{J}_1(\phi; \mu_n, \nu_n)\mathbf{1}_{\mathcal{B}} - \mathcal{J}_1(\phi; \mu_n, \nu_n)\mathbf{1}_{\mathcal{B}^c}], \tag{3.8}$$

where $\mathbf{1}_A(x) = 1$ if $x \in A$ and $\mathbf{1}_A(x) = 0$ if $x \in A^c$ for a given set $A$.

**Theorem 3.** *Given* $\mu, \nu \in \mathcal{P}(\Omega)$*, the optimal value of (3.8) corresponds to* $W_1(\mu, \nu)$*.*

*Proof.* From Theorem 1, (3.7) gives the Wasserstein distance between $\mu$ and $\nu$, that is,

$$W_1(\mu, \nu) = \sup_{\phi \in \mathcal{B}} \left\{ \mathbb{E}_{\substack{X_i \sim \mu, Y_i \sim \nu \\ 1 \leq i \leq n}}[\mathcal{J}_1(\phi; \mu_n, \nu_n)] \right\}. \tag{3.9}$$

Let $\phi$ be the maximizer of the equation above. Since

$$\mathcal{J}_1(\phi; \mu_n, \nu_n) \leq \mathcal{J}_2(\phi; \mu_n, \nu_n) \quad \text{and} \quad \mathcal{J}_1(\phi; \mu_n, \nu_n) \leq \mathcal{J}_3(\phi; \mu_n, \nu_n),$$

we have that (3.8) is greater than or equal to (3.9).

The converse is true thanks to the condition $\mathcal{J}_1(\phi; \mu_n, \nu_n) \geq 0$ in (3.8) and we finish the proof. $\square$

# 4 Algorithm

## 4.1 Algorithm for estimating the Wasserstein distance

Inspired from (3.8), we propose an iterative scheme for estimating the Wasserstein distance between two probability measures, which is given in Algorithm 1. Let us illustrate its procedure in detail. We begin by parameterizing $\phi$ by $\eta$, such as deep neural networks with a parameter $\eta$. Given two sets of i.i.d samples $X_i \sim \mu$ and $Y_i \sim \nu$ for $i = 1, ..., n$, one computes $\mathcal{J}_k(\phi; \mu_n, \nu_n)$ for $k \in \{1, 2, 3, 4\}$. If $\mathcal{J}_1(\phi; \mu_n, \nu_n)$ is greater than either $\mathcal{J}_2(\phi; \mu_n, \nu_n)$ or $\mathcal{J}_3(\phi; \mu_n, \nu_n)$, one updates $\eta$ do increase $\mathcal{J}_1(\phi; \mu_n, \nu_n)$. Otherwise, $\eta$ is updated to decrease $\mathcal{J}_1(\phi; \mu_n, \nu_n)$.

This way, we can build a 1-Lipschitz continuous function $\phi$ that better approximates the Wasserstein distance between probability measures using mini-batches. Note that Adam optimizer (Kingma & Ba, 2014) is used for updating neural network parameters throughout the paper.

---
**Algorithm 1** Our proposed algorithm of estimating Wasserstein distance
---
**for** *iter of training iterations* **do**
    **if** $\mathcal{J}_2(\cdot; \mu_n, \nu_n) < \mathcal{J}_1(\cdot; \mu_n, \nu_n)$ *or* $\mathcal{J}_3(\cdot; \mu_n, \nu_n) < \mathcal{J}_1(\cdot, \mu_n, \nu_n)$ **then**
        $\eta \leftarrow \text{Adam}(\mathcal{J}_1(\cdot, \mu_n, \nu_n), \eta)$
    **end**
    $\eta \leftarrow \text{Adam}(-\mathcal{J}_1(\cdot, \mu_n, \nu_n), \eta)$
**end**

---

## 4.2 Main Algorithm for WGAN

We now propose a new framework for learning a target distribution $\nu \in \mathcal{P}$ by solving

$$\inf_{\nu \in P(\Omega)} \sup \left\{ \int_\Omega \phi \mathrm{d}(\mu - \nu) : \phi \text{ satisfies } (3.1) \right\}. \tag{4.1}$$

Based on the observation in Lemmas 1 and 2, we propose the following training procedure as described in Algorithm 2. For the discriminator $\phi$ and the generator parametrized by $\eta$ and $\theta$ respectively, the algorithm begins with minimizing $\mathcal{J}_1(\phi; \mu_n, \nu_n)$ if either $\mathcal{J}_2(\phi; \mu_n, \nu_n) < \mathcal{J}_1(\phi; \mu_n, \nu_n)$ or $\mathcal{J}_3(\phi; \mu_n, \nu_n) < \mathcal{J}_1(\phi; \mu_n, \nu_n)$

holds. Otherwise, $\phi$ is updated to maximize $\mathcal{J}_1$. For the generator step, we follow the standard scheme. The generator parameter $\theta$ is updated to minimize $\mathcal{J}_1(\phi; \mu_n, \nu_n)$ while $\phi$ fixed. Since the algorithm depends on the comparison between $\mathcal{J}_i$'s, we call it **CoWGAN**. A salient feature of this new algorithm is that it only requires that $\phi$ be Lipschitz indirectly through the comparison process while keeping the generator-update step the same as classical algorithms such as WGAN-GP.

---

**Algorithm 2** Our proposed algorithm (**CoWGAN**) to find the minimum of (4.1)

---

**for** *iter of training iterations* **do**
    **for** $t = 1, 2, \ldots, n_{critic}$ **do**
        **if** $\mathcal{J}_2(\cdot; \mu_n, \nu_n) < \mathcal{J}_1(\cdot; \mu_n, \nu_n)$ *or* $\mathcal{J}_3(\cdot; \mu_n, \nu_n) < \mathcal{J}_1(\cdot; \mu_n, \nu_n)$ **then**
            $\eta \leftarrow \mathrm{Adam}(\mathcal{J}_1(\cdot; \mu_n, \nu_n), \eta)$
        **end**
        $\eta \leftarrow \mathrm{Adam}(-\mathcal{J}_1(\cdot; \mu_n, \nu_n), \eta)$
    **end**
    $\theta \leftarrow \mathrm{Adam}(\mathcal{J}_1(\cdot; \mu_n, \nu_n), \theta)$
**end**

---

### 4.3 Why not optimizing $\mathcal{J}_2$ or other objective functions?

In this section, we provide an illustrative example that empirically justifies our approach.

In practice, one does not have access to the true distribution but rather to mini-batches that are sampled from the training dataset available. Although we expect $\mathcal{J}_1$ to be close to $\mathcal{J}_2$ and $\mathcal{J}_3$, we observe that the objective functions $\mathcal{J}_2$ and $\mathcal{J}_3$ may not be suitable for constructing the true distribution. As shown in Fig. 2, blurry images are achieved if $\mathcal{J}_2$ is used as an objective function.

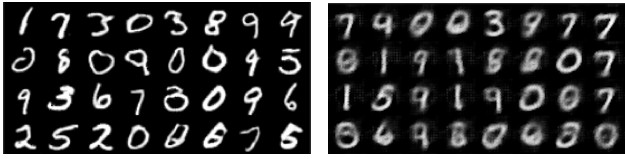

Figure 2: In comparison to CoWGAN (left), only $J_2$ is used in *c*-transform (right)

In what follows, we discuss why we do not use $\mathcal{J}_2$ or $\mathcal{J}_3$ as objective functions but rather use $\mathcal{J}_1$, and why other methods based on $\mathcal{J}_2$ may lead to poor generators.

The problem of minimizing (2.4) with respect to $\nu$ cannot be solved by considering $\mathcal{J}_2$ or $\mathcal{J}_3$ instead of $\mathcal{J}_1$:

$$\inf_{\nu \in P(\Omega)} \sup_{\phi \in \mathrm{Lip}(\Omega)} \mathbb{E}_{\substack{X_i \sim \mu, Y_i \sim \nu \\ 1 \leq i \leq n}} \left[ \mathcal{J}_2 \text{ (or } \mathcal{J}_3) \right]. \tag{4.2}$$

We note from (3.5) that

$$\mathcal{J}_2 = \frac{1}{n} \sum_{i=1}^{n} \left( \phi(X_i) - \min_{1 \leq j \leq n} \{ \|X_j - Y_i\| - \phi(X_j) \} \right), \tag{4.3}$$

and $\mathbb{E}[\mathcal{J}_1] \neq \mathbb{E}[\mathcal{J}_i]$ for $i = 2, 3$ in general.

We will demonstrate that replacing $\mathcal{J}_1$ with $\mathcal{J}_2$ might lead to a poor estimation of the target measure $\mu$. To see this, setting $\mu = \mu_m$ for some $m > 1$ and $n = 1$ in (4.2) with an objective function $\mathcal{J}_2$, (4.3) reduces to

$$\inf_{\nu \in \mathcal{P}(\Omega)} \mathbb{E}_{X \sim \mu_m, Y \sim \nu} \|X - Y\|. \tag{4.4}$$

The question is whether an optimal $\nu$ in (4.4) approximates the given probability measure $\mu$. The answer is *no* as illustrated in the following proposition and Fig. 2, which justifies the use of $\mathcal{J}_1$ as an objective function in our algorithm.

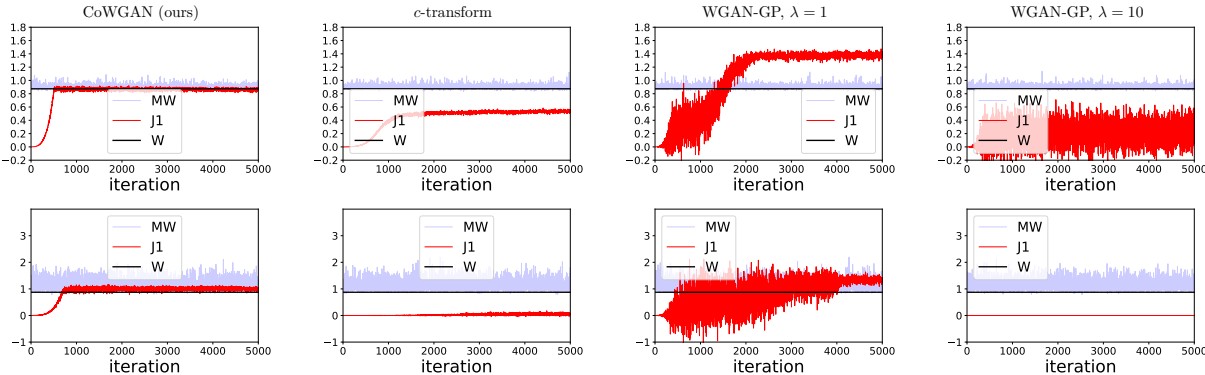

Figure 3: $\mathcal{J}_1$, the true Wasserstein distance (W), and the minibatch Wasserstein distance (MW) with mini-batch size 64 (top) and 8 (bottom) based on two 4-Gaussian synthetic datasets. Optimal Transport Toolbox (Flamary et al., 2021) is used for comparison.

**Proposition 2.** *If $\Omega \subset \mathbb{R}$, the minibatch size $n = 1$ and $\mu \in \mathcal{P}_m(\Omega)$ for $m > 1$, then $\nu = \delta_y$ is a global minimizer of (4.4) for any median $y$ of $\mu$ where $\mathcal{P}_m(\Omega)$ denote a set of empirical measures supported on at most $m$ points in $\Omega$. The median of a probability measure $\mu$ is defined as a real number $k$ satisfying $\mu((-\infty, k]) \geq \frac{1}{2}$ and $\mu([k, \infty)) \geq \frac{1}{2}$. One can achieve multiple medians when $\mathrm{supp}(\mu)$ contains intervals.*

*Proof.* To see this let us define an empirical measure $\mu_m = \frac{1}{m} \sum_{i=1}^m \delta_{X_i}$ with $X_1 < X_2 < \cdots < X_m$.

We claim that if $y \in [X_{[(m+1)/2]}, X_{[(m+2)/2]}]$, then $\nu = \delta_y$ is a global minimizer of (4.4). As $d = n = 1$, the objective function in (4.4) can be represented as

$$\mathbb{E}_{X \sim \mu_m, Y \sim \nu} \left[ \|X - Y\| \right] =: I.$$

To proceed, by the triangular inequality,

$$I = \frac{1}{2} \mathbb{E}_{Y \sim \nu} \left[ \sum_{i=1}^m \|X_i - Y\| + \|X_{m+1-i} - Y\| \right]$$

$$\geq \frac{1}{2} \sum_{i=1}^m \|X_i - X_{m+1-i}\|.$$

On the other hand, if $\nu = \delta_y$ for $y \in [X_{[(m+1)/2]}, X_{[(m+2)/2]}]$, then we have

$$\mathbb{E}_{Y \sim \nu} \left[ \sum_{i=1}^m \|X_i - Y\| \right] = \sum_{i=1}^{[(m+1)/2]} (y - X_i) + \sum_{i=[(m+2)/2]}^m (X_i - y)$$

$$= \sum_{i=1}^{[(m+1)/2]} (X_{m+1-i} - X_i).$$

As the minimum of (4.4) is attained at $\nu = \delta_y$, we finish the proof. $\qquad\square$

The result implies that $\nu$ might completely fail to mimic $\mu$ using mini-batches when $\mu$ is a discrete measure. This observation coincides with the blurry images obtained in previous works using the $c$-transform method to train WGANs (Mallasto et al., 2019) and discussions presented in Stanczuk et al. (2021).

In general, the discrepancy between the true distance and the estimated distance using finite samples has been studied by Arora et al. (2017; 2018); Bai et al. (2018). Furthermore, in higher dimensions, even if both $\mu_n, \nu_n$ come from the same distribution $\mu$, one has $W_1(\mu_n, \nu_n) \geq C > 0$ for some $C$; see the works of Fatras et al. (2020; 2021) for details.

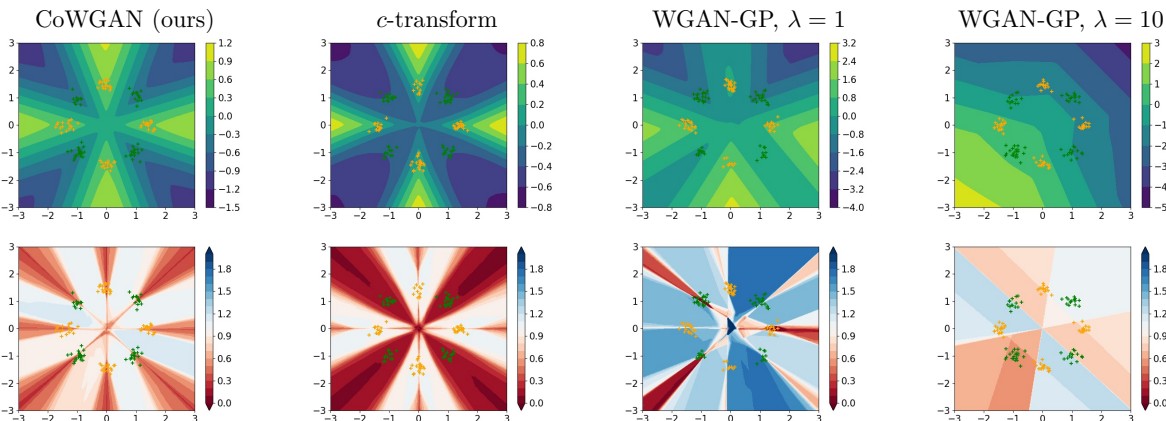

Figure 4: The discriminator $\phi$ for two mixtures of 4 Gaussians (samples shown as green and yellow dots) after 2000 iterations with different methods and mini-batch size 64 (top). The gradient norm of the discriminator $\|D\phi\|$ for two mixtures of 4 Gaussians after 2000 iterations with different methods and mini-batch size 64 (bottom).

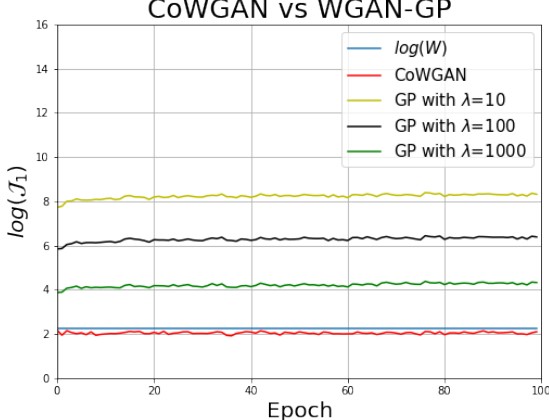

Figure 5: $\log(\mathcal{J}_1)$, and the logarithm of the true Wasserstein distance $\log(W)$ between 5,000 images of digit 1 and 5,000 images of digit 2 from the MNIST dataset

## 5    Experiments

Various empirical results are provided to validate our method. We first test our algorithm on various training sets ranging from Gaussian mixture distribution to the MNIST dataset to estimate the 1-Wasserstein distance. In the subsequent part, we validate the effectiveness of the proposed algorithm in the generative setting. Particularly, we focus on the enforcement of the 1-Lipschitz condition, the landscape of loss during the training, and the effect of hyperparameters.

### 5.1    Estimating the 1-Wasserstein distance

In this section, we provide empirical demonstrations of how well our method computes the 1-Wasserstein distance using Algorithm 1. Synthetic data, as well as various benchmark datasets such as MNIST, Fashion-MNIST, CIFAR10, and CelebA are used for the experiments. Furthermore, we also show the effectiveness of our learning algorithm for estimating the Wasserstein distance even with a small batch size.

### 5.1.1 Synthetic datasets in 2D

On synthetic datasets in 2D, we observe that the condition (3.1) can be effectively enforced when using the comparison method.

In Fig. 3, we compute the Wasserstein distance between two different distributions based on four different methods. Each distribution is a mixture of 4 different Gaussian distributions. Samples from them are denoted by yellow and green points as shown in Fig. 4. It is observed that CoWGAN converges much faster than the other algorithms tested, namely the $c$-transform method (Mallasto et al., 2019), and WGAN-GP with $\lambda = 1$ or $\lambda = 10$ (Gulrajani et al., 2017) as shown in Fig. 3.

The discriminator $\phi$ obtained by Algorithm 1 captures the feature that distinguishes two distinct distributions with the same training time, which is the desired property for $\phi$. It is demonstrated in Fig. 4. In particular, we observe that in our algorithm $\|D\phi\| = 1$ a.e. and the Lipschitzness of $\phi$ is enforced well.

The optimal discriminator should have level sets orthogonal to the transport map. As shown in Fig. 4, this property is observed in the result obtained by our method, unlike others.

### 5.1.2 The MNIST dataset

Computing the Wasserstein distance for high-dimensional data is challenging in general. We experimentally verify that CoWGAN efficiently can estimate the Wasserstein distance between probability measures on a high-dimensionality.

In Fig. 5, we sampled 5,000 images of digit 1 and 5,000 images of digit 2 from the MNIST dataset. Applying Algorithm 1, we compute the Wasserstein distance between the two distributions, one representing 1 and the other representing 2. We then compare the outcome of the algorithm with the true Wasserstein distance between two sets of 5,000 images computed using the POT Python Library (Flamary et al., 2021). We note that the algorithm well approximates the distance well while others induce non-negligible bias.

### 5.1.3 Training with small mini-batches

Surprisingly, Algorithm 1 can enforce the Lipschitz constraint very effectively, even with a very small mini-batch size. We consider the same distributions as in the previous experiment, Fig. 4, but with the reduced size of the mini-batches, 8. The result is presented in Fig. 6, and it shows that our algorithm still accurately computes the optimal discriminator, yet the other methods have significant difficulties even after more iterations. It is quite natural to see such a phenomenon as the gradient penalty method tries to enforce the norm of the gradient of a randomly chosen point between real and synthetic data to 1. On the other hand, CoWGAN just imposes a condition on the slope between the data, real and synthetic. As a consequence, the norm of the slope is maintained to be a value close to 1.

### 5.2 Learning generative models

We now apply our algorithm to generative learning in a framework of WGAN. Several experimental results using the MNIST, Fashion MNIST, CIFAR-10, as well as CelebA-HQ datasets resized to $64 \times 64$ and $128 \times 128$ are presented., Again we utilize the algorithm proposed in the previous section to generate synthetic images.

We find that the training procedure is more stable compared with WGAN-GP in terms of the Lipschitz estimate while generating images of similar quality. In particular, the proposed machinery is robust to hyperparameter tuning as the algorithm does not involve any parameters such as weight $\lambda$ or $n_{critic}$, which represents the number of discriminators updated per each generator update. The details on the used neural network architecture are provided in Appendix A.

### 5.2.1 1-Lipschitzness

We present the quantitative differences between our algorithm and the classical WGAN-GP by examining the Lipschitz constant computed with samples. To this end, we first sample 64 points from each real and synthetic

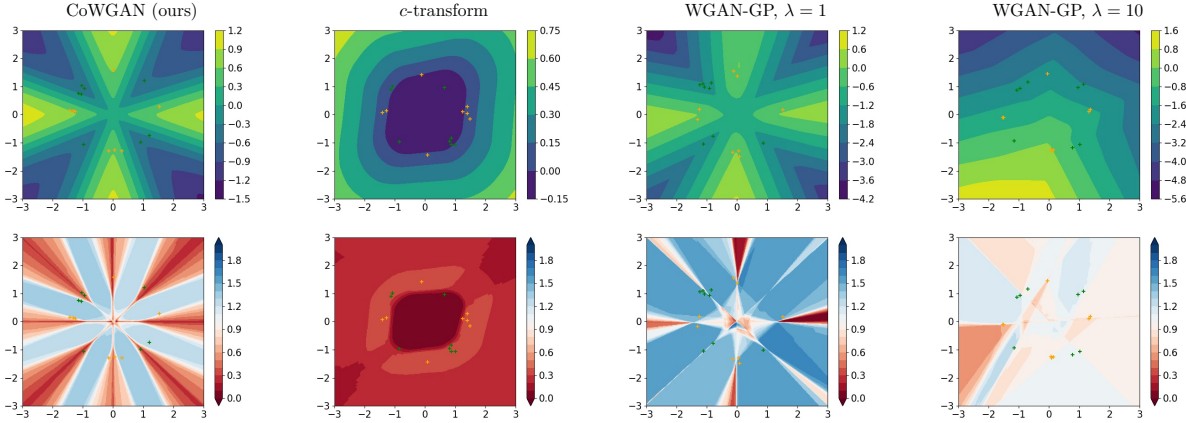

Figure 6: The discriminator $\phi$ after 5,000 iterations with mini-batches of size 8 (top). Shown is $\|D\phi\|$ after 5,000 iterations with mini-batches of size 8 (bottom).

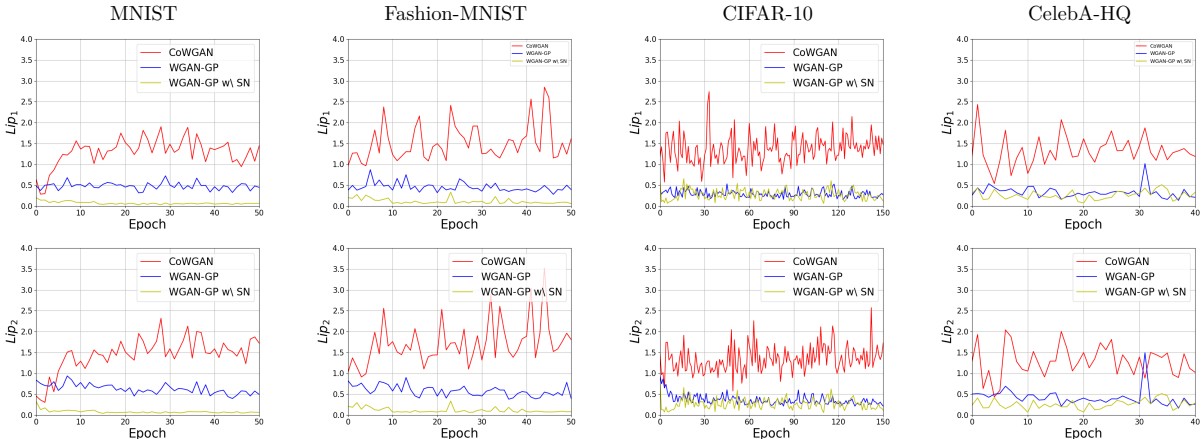

Figure 7: Lipschitz estimate within the real data denoted by $\text{Lip}_1$ (top), and synthetic data denoted by $\text{Lip}_2$ (bottom); Lipschitz constant is close to 1 in our algorithm while WGAN-GP underestimates it.

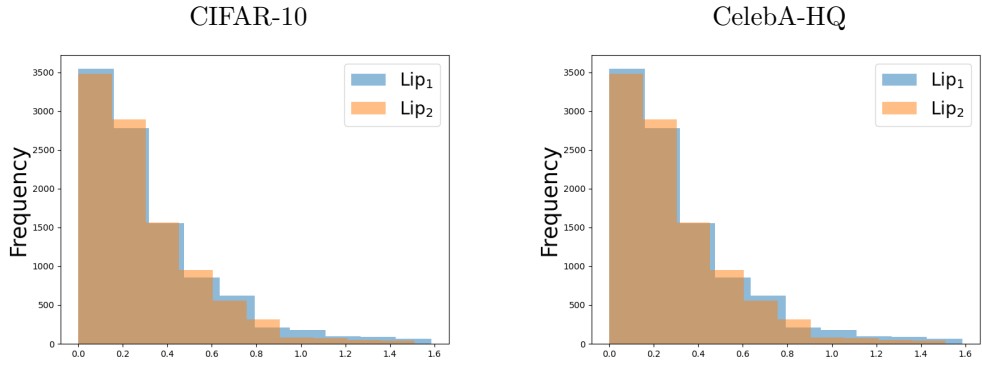

Figure 8: Empirical density of Lipschitz constant computed with 10,000 pairs of real/synthetic dataset.

dataset and compute the maximum slope of the discriminator within them, i.e., $\sup_{x \sim \mu, y \sim \nu} \left| \frac{\phi(x) - \phi(y)}{\|x - y\|} \right|$ and $\sup_{x, y \sim \mu} \left| \frac{\phi(x) - \phi(y)}{\|x - y\|} \right|$.

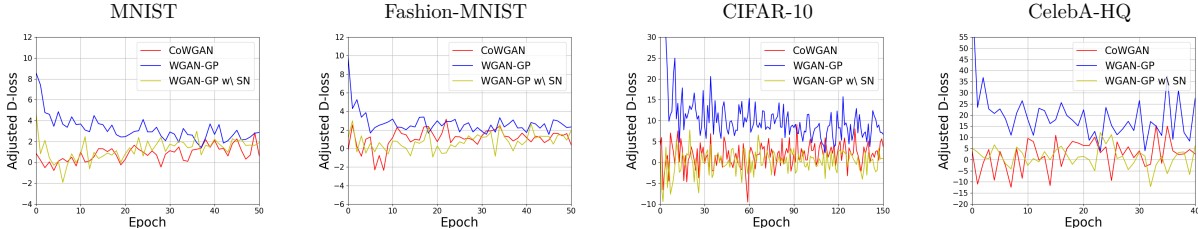

Figure 9: The adjusted discriminator loss (the discriminator loss/the Lipschitz norm). Due to the smaller Lipschitz estimate in Fig. 7, WGAN-GP underestimates the loss function. Our losses are smaller when the Lipschitz norms of each discriminator are normalized to one.

The result is shown in Fig. 7. We observe that WGAN-GP and WGAN-GP with spectral normalization (SN) constantly underestimate the Lipschitz constant. On the other hand, our algorithm maintains the Lipschitz constant slightly above 1. This is mainly because the gradient penalty term unnecessarily forces the gradient of points to be 1, even if the norm of their gradients does not have to be equal to one. Precisely, the optimal $\phi$ satisfies $\|D\phi\| = 1$ only along the optimal coupling (see, e.g., Proposition 1 in Gulrajani et al. (2017)), but the WGAN-GP algorithm imposes this condition for random points between real and synthetic data which are chosen arbitrarily. However, our algorithm searches for a 1-Lipschitz continuous discriminator directly by making use of the necessary condition, the duality.

In addition, Fig. 8 shows the empirical density of the Lipschitz constant, calculated using 10,000 pairs of data from both real and synthetic datasets. These histograms provide a more detailed view of how the Lipschitz constant behaves in practice. As illustrated, for more than 98.5% of the data pairs, the Lipschitz constant remains less than or equal to 1, demonstrating that our method enforces the 1-Lipschitz condition well for the majority of the data.

### 5.2.2 Loss and normalized loss – comparison after normalization

As shown in Fig. 7, WGAN-GP consistently underestimates the Lipschitz condition tested for real and synthetic data, which does not agree with the intuition behind WGAN-GP; the optimal discriminator is 1-Lipschitz continuous, and the norm of gradients of the optimal discriminator is equal to 1 along arbitrary optimal pairs.

We point out that the discriminator trained using the gradient penalty yields the possible upper bound for the 1-Wasserstein distance between two distributions rather than the actual Wasserstein distance.

**Remark 2.** *Let $\nu_{gp}$ and $\nu_{co}$ be the distribution measures generated by WGAN-GP and CoWGAN, respectively. As in Fig. 9, we empirically observe that the normalized loss function of CoWGAN is smaller than that of WGAP-GP:*

$$\|\phi_{co}\|_{\mathrm{Lip}}^{-1}\mathcal{J}_1(\phi_{co};\mu,\nu_{co}) < \|\phi_{gp}\|_{\mathrm{Lip}}^{-1}\mathcal{J}_1(\phi_{gp};\mu,\nu_{gp}),$$

*where $\|f\|_{\mathrm{Lip}}$ denotes the Lipschitz constant of $f$. Given Theorem 3, our method computes the 1-Wasserstein distance, at least in theory. Therefore, we conclude that our distribution is closer to the true distribution $\mu$:*

$$W_1(\mu,\nu_{co}) < W_1(\mu,\nu_{gp}). \tag{5.1}$$

Hence, the effectiveness of our method for estimating the 1-Wasserstein distance is corroborated by this empirical evidence, as our normalized loss always lies below that of WGAN-GP.

### 5.2.3 Effect of $n_{critic}$

WGAN-GP and many subsequential improved algorithms usually choose $n_{critic}$ to be five when $d = g = 0.0001$, which allows the algorithm to train the discriminator enough before updating the generator. In this subsection, we heuristically analyze the effect of $n_{critic}$ in our algorithm by choosing $n_{critic} \in \{1, 5, 10\}$. For

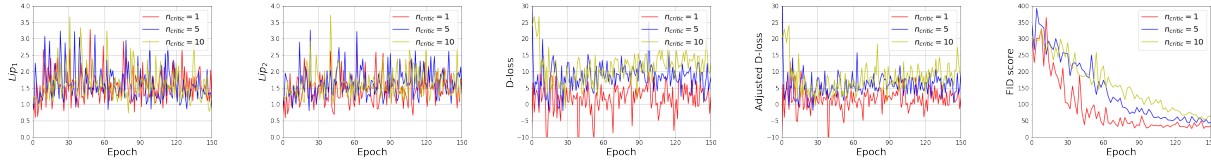

Figure 10: With the same learning rate for both discriminator and generator, $d = g = 0.0001$, the effect of $n_{critic}$ that indicates the number of discriminator updates per generator update is demonstrated using the CIFAR-10 dataset.

the learning rate, we use $d = g = 0.0001$. As seen in Fig. 10, $n_{critic} = 1$ shows the best performance in terms of the stability of the loss curve and quality of synthetic pictures generated. Hence, we deduce that it is not necessary to train the discriminator more than the generator under the CoWGAN framework.

### 5.2.4 Hyperparameter tunning

As noted, our algorithm does not contain the penalizing term, which is usually denoted by $\lambda$, unlike most WGAN-GP and related models built on it, such as Wei et al. (2018).

On the other hand, the choice of learning rate is a bit delicate. Denoting the learning rate for the discriminator and generator by $d$ and $g$ respectively, it is common to set $d = g = 0.0001$. Meanwhile, Heusel et al. (2017) evaluates the effectiveness of implementing a two-time scale update for training GANs. Motivated by this, we set $g > d$, and the ratio between $g$ and $d$ we use for the experiments is given in Table 1.

|  | Dataset | Discriminator (d) | Generator (g) |
|---|---|---|---|
| CoWGAN | MNIST,FMNIST | $d = 0.0001$ | $5d$ |
|  | CIFAR-10 | $d = 0.0001$ | $2d$ |
|  | CelebA-HQ | $d = 0.00005$ | $2d$ |
| WGAN-GP | same for all data | $d = 0.0001$ | $d$ |

Table 1: Diffrent learning rate depending on the training data.

### 5.2.5 Performance of generator

| Dataset | Method | Inception Score (IS) ↑ | Frechet Inception Distance (FID) ↓ |
|---|---|---|---|
| CIFAR10 | CoWGAN | $6.03 \pm 0.13$ | 26.34 |
|  | WGAN-GP | $6.14 \pm 0.16$ | 27.66 |
|  | WGAN-GP w/ SN | $5.05 \pm 0.10$ | 42.42 |
| CelebA-HQ | CoWGAN | $2.98 \pm 0.05$ | 19.42 |
|  | WGAN-GP | $2.97 \pm 0.21$ | 16.74 |
|  | WGAN-GP w/ SN | $2.87 \pm 0.13$ | 26.32 |

Table 2: Comparison of CoWGAN and WGAN-GP on CIFAR and CelebA Datasets

Our algorithm performs well for generating synthetic images without computing the gradients explicitly, as can be seen in Fig. 11 and Table 2. We test our algorithm with various datasets ranging from MNIST, Fashion-MNIST, and CIFAR-10, which are resized to $32 \times 32$, and CelebA, which is resized to $64 \times 64$. An additional experiment with the CelebA-HQ dataset resized into $128 \times 128$ was conducted, and the result is presented in Appendix B. Our algorithm and WGAN-GP generate comparable images. However, when using the same neural network architecture (See Appendix A), WGAN-GP with spectral normalization exhibits a slightly lower IS (Inception Score) and higher FID (Frechet Inception Distance). This lower performance

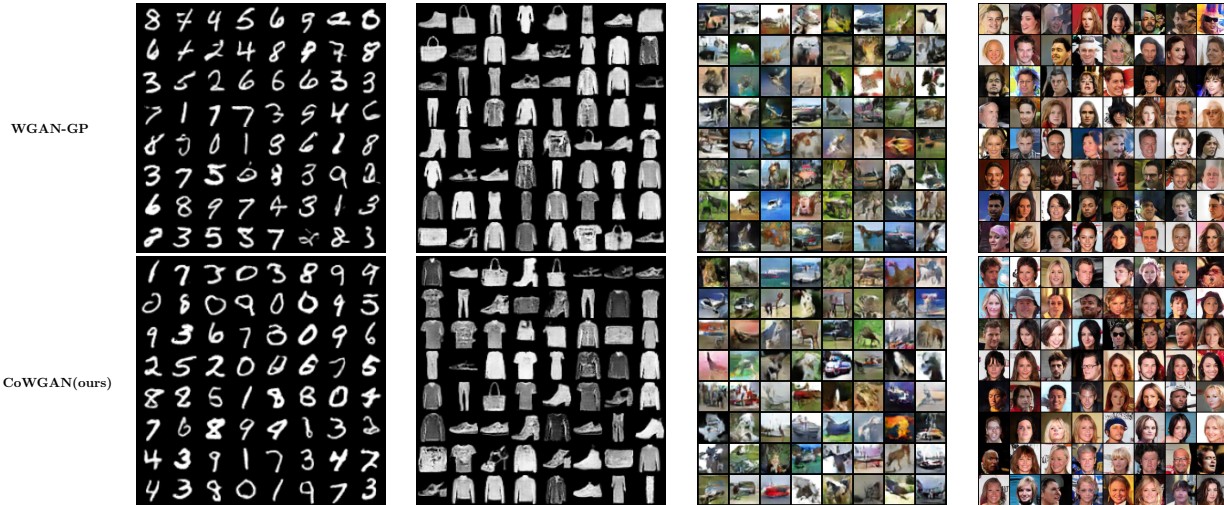

Figure 11: Top line is generated by WGAN-GP; the bottom is generated via CoWGAN. From left to right, MNIST, Fashion-MNIST, CIFAR-10, and CelebA-HQ, that is resized to $64 \times 64$ are used. Visually, the generated images are of similar quality, but our algorithm runs six times faster in wall-clock time. The batch size is chosen to be 64.

has been discussed in Miyato et al. (2018), where authors note that increasing the number of feature maps can negatively impact the performance of WGAN-GP with spectral normalization.

# 6 Conclusion

In this work, we propose a novel algorithm that accurately estimates the Wasserstein distance between two probability measures and can be used to train generative models. We point out that enforcing the gradient norm condition tends to underestimate the Wasserstein distance. The proposed algorithm exploits a combination of objective functions inspired by the recently proposed back-and-forth method. It is supported by sound theoretical properties obtained by Kantorovich duality and a few new inequalities. Importantly, our method does not require an explicit computation of the gradients of the discriminator to implement the Lipschitz constraint on the optimal discriminators in WGANs. A salient feature of our method is the absence of gradient penalties, eliminating the need for corresponding hyperparameter tuning. The effectiveness of this approach has been verified through numerous experiments. We also highlight that adding penalization in the cost function, which is usually intended to force unknown functions to have desirable properties, needs a careful examination when applied.

## Broader Impact Statement

There are many potential societal consequences of our work, but we do not believe any of them need to be specifically highlighted here.

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

## Appendix

## A    Neural Network Architecture

We use standard convolution neural network (CNN) architecture for the experiments as in Gulrajani et al. (2017); Wei et al. (2018). For MNIST and Fashion MNIST, the sigmoid function is utilized for the final activation in the generator, while Tanh is used for color images, CIFAR-10, and CelebA-HQ.

| Discriminator |
| :---: |
| Input: 1*32*32 |
| 4*4 conv. 256, Pad = 1, Stride = 2, lReLU 0.2 |
| 4*4 conv. 512, Pad = 1, Stride = 2, lReLU 0.2 |
| 4*4 conv. 1024, Pad = 1, Stride = 2, lReLU 0.2 |
| 4*4 conv. 1, Pad = 0, Stride = 1 |
| Reshape 1024*4*4 (D) |
| Sigmoid |
| **Generator** |
| Input: Noise z 100 |
| 4*4 deconv. 1024, Pad = 0, Stride = 1, BatchNorm2d, ReLU |
| 4*4 deconv. 512, Pad = 1, Stride = 2, BatchNorm2d, ReLU |
| 4*4 deconv. 256, Pad = 1, Stride = 2, BatchNorm2d, ReLU |
| 4*4 deconv. 1, Pad = 1, Stride = 2, BatchNorm2d, ReLU |

Table 3: Neural network Architectures for MNIST and Fashion-MNIST

| Discriminator |
| :---: |
| Input: 1*32*32 |
| 4*4 conv. 256, Pad = 1, Stride = 2, lReLU 0.2 |
| 4*4 conv. 512, Pad = 1, Stride = 2, lReLU 0.2 |
| 4*4 conv. 1024, Pad = 1, Stride = 2, lReLU 0.2 |
| 4*4 conv. 1, Pad = 0, Stride = 1 |
| Reshape 1024*4*4 (D) |
| Tanh |
| **Generator** |
| Input: Noise z 100 |
| 4*4 deconv. 1024, Pad = 0, Stride = 1, BatchNorm2d, ReLU |
| 4*4 deconv. 512, Pad = 1, Stride = 2, BatchNorm2d, ReLU |
| 4*4 deconv. 256, Pad = 1, Stride = 2, BatchNorm2d, ReLU |
| 4*4 deconv. 1, Pad = 1, Stride = 2, BatchNorm2d, ReLU |

Table 4: Neural network Architectures for CIFAR-10

| Discriminator |
| --- |
| Input: 3*128*128 |
| 4*4 conv. 256, Pad = 1, Stride = 2, lReLU 0.2 |
| 4*4 conv. 512, Pad = 1, Stride = 2, lReLU 0.2 |
| 4*4 conv. 1024, Pad = 1, Stride = 2, lReLU 0.2 |
| 4*4 conv. 2048, Pad = 1, Stride = 2, lReLU 0.2 |
| 4*4 conv. 4096, Pad = 1, Stride = 2, lReLU 0.2 |
| 4*4 conv. 1, Pad = 0, Stride = 1 |
| Reshape 4096*4*4 (D) |
| Tanh |
| **Generator** |
| Input: Noise z 100 |
| 4*4 deconv. 4096, Pad = 0, Stride = 2, BatchNorm2d, ReLU |
| 4*4 deconv. 2048, Pad = 1, Stride = 2, BatchNorm2d, ReLU |
| 4*4 deconv. 1024, Pad = 1, Stride = 2, BatchNorm2d, ReLU |
| 4*4 deconv. 512, Pad = 1, Stride = 2, BatchNorm2d, ReLU |
| 4*4 deconv. 256, Pad = 1, Stride = 2, BatchNorm2d, ReLU |
| 4*4 deconv. 1, Pad = 1, Stride = 2, BatchNorm2d, ReLU |

Table 6: Neural network Architectures for CelebA-HQ ($128 \times 128 \times 3$)

| Discriminator |
| --- |
| Input: 3*64*64 |
| 4*4 conv. 256, Pad = 1, Stride = 2, lReLU 0.2 |
| 4*4 conv. 512, Pad = 1, Stride = 2, lReLU 0.2 |
| 4*4 conv. 1024, Pad = 1, Stride = 2, lReLU 0.2 |
| 4*4 conv. 2048, Pad = 1, Stride = 2, lReLU 0.2 |
| 4*4 conv. 1, Pad = 0, Stride = 1 |
| Reshape 2048*4*4 (D) |
| Tanh |
| **Generator** |
| Input: Noise z 100 |
| 4*4 deconv. 2048, Pad = 0, Stride = 2, BatchNorm2d, ReLU |
| 4*4 deconv. 1024, Pad = 1, Stride = 2, BatchNorm2d, ReLU |
| 4*4 deconv. 512, Pad = 1, Stride = 2, BatchNorm2d, ReLU |
| 4*4 deconv. 256, Pad = 1, Stride = 2, BatchNorm2d, ReLU |
| 4*4 deconv. 1, Pad = 1, Stride = 2, BatchNorm2d, ReLU |

Table 5: Neural network Architectures for CelebA-HQ ($64 \times 64 \times 3$)

## B    Additional experiments

We also test that CoWGAN performs well in a generative setting with high-dimensional data, center-cropped $128 \times 128$ CelebA-HQ, which is shown in Fig. 12. The learning rate is set $g = 3d$ with $d = 0.00001$ where $d$ denotes the learning rate of the discriminator, and the neural network architecture is described in Table 6.

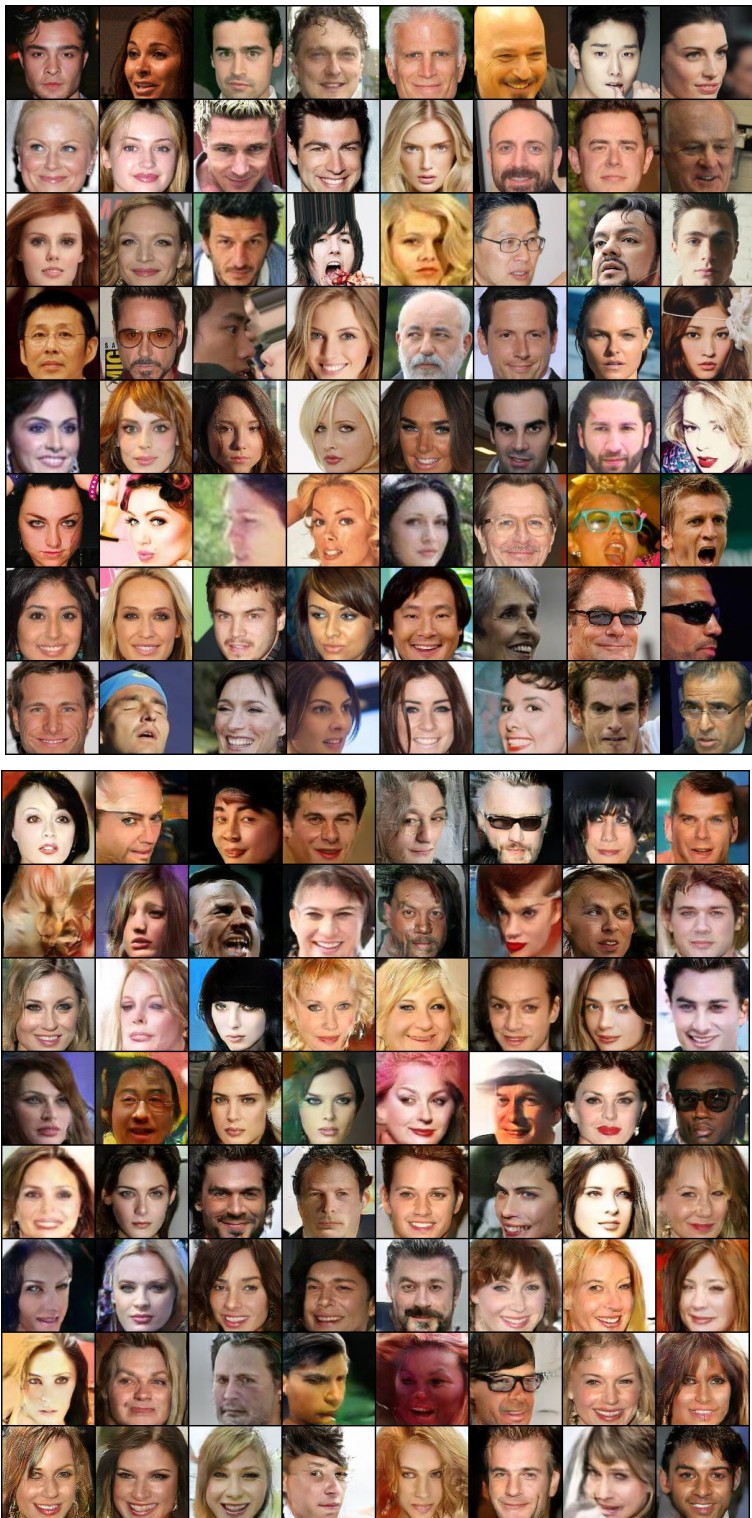

Figure 12: Real (top) and synthetic (bottom) data generated by CoWGAN algorithm using $128 \times 128$ center-cropped CelebA dataset.

