# OpenReview forum: "Training Wasserstein GANs without Gradient Penalties via Duality"
_TMLR — Rejected by TMLR_

### Review · Reviewer_u9Aw · 2024-07-21

**Summary Of Contributions:**

The paper proposed an iterative algorithm for estimating Wasserstein distance from samples, which can be used for training Wasserstein Generative Adversarial Networks (WGANs). Unlike prior approaches, which commonly use gradient penalties to enforce Lipschitz constraints on the discriminator, the proposed method, named CoWGAN, does not rely on such penalties; instead, it leverages the c-transform based on Kantorovich duality to formulate new objective functions that more accurately estimate the Wasserstein distance between datasets. This method ensures the discriminator adheres to the 1-Lipschitz constraint, reducing the need for gradient penalties and the associated hyperparameter tuning. Theoretical analysis is provided to justify the correctness the algorithm and also demonstrate issues with other alternatives. Through extensive experiments, the authors claim that their method provides an improved Wasserstein distance estimator and discriminator, leading to high-quality generated images across various datasets, including MNIST, Fashion-MNIST, CIFAR-10, and CelebA-HQ. The results also demonstrates, on a high level, that directly incorporating penalties to the cost function may have undesired consequences.

**Audience:**

Yes

**Broader Impact Concerns:**

I don't see additional concern here.

**Claims And Evidence:**

No

**Requested Changes:**

- [Minor] Proposition 1 is trivial and well known. A reference on it should suffice, with only minimal discussion necessary.

- [Critical] “Accuracy” of $\mathcal{J}_1$ as an estimator is frequently brought up (Section 3), yet I cannot find much concrete and precise support on it and, in fact, I am not sure what “accurate” means mathematically. To me, Theorem 2 suggests that $\mathcal{J}_1$ could be a reasonable estimate that is potentially unbiased (under conditions on $\phi$). To make such claims, I would expect some more rigorous argument that provides quantification of errors.
- [Minor] Related, the top paragraph on Page 9 (above Section 4) seems vague and imprecise. For instance, the mode of convergence must be specified in the centered equation.
- [Important, though not critical] I am not sure of the significance of Proposition 2. To me, it shows the potential non-uniqueness of the optimal solution, yet this does not directly address the question of whether it “is similar to” the desired measure. This is a reasonable attempt at providing intuitive justification, but I just do not think it truly serves the purpose well.
- [Minor, though important in aggregation] Typos and readability:
    - Citation styles: Narrative citations and parenthetical citations should be distinguished in their uses. For instance, the very first sentence of the introduction should read “Generative Adversarial Networks (GANs) (Goodfellow et al., 2014) have …”. The same applies to equation references.
    - The centered equation below Eqn (1.1): the second $\sup$ should have $\phi\in C(\Omega)$ below it.
    - Below Eqn (WGAN-GP): “A central to” - a typo or runaway statement?
    - Section 4.1, Line 5: “either” mis-spelled
    - Page 10, last line: “to have $\mathcal{J}_1$ is close to …” - please fix the grammar

**Strengths And Weaknesses:**

Overall, the paper is stronger on the empirical side and weaker on the theoretical side. The presentation of the content also requires additional polishing

[Strengths]
- The algorithm for estimating Wasserstein distances and training WGANs is novel, in particular in its elimination of the penalty term. Extending prior works, it provides a reconsideration of how to deal with the Wasserstein constraint in WGANs.
- The empirical evaluation is thorough and extensive on both synthetic and real datasets. The results well demonstrate the effectiveness of the CoWGAN algorithm.

[Weaknesses]
- On the theoretical side, I find several of the lemmas/propositions may have their significance over-stated. For instance, I believe Proposition 1, the simplest case of Kantorovich-Rubenstein duality, is well known, and the proof and discussion on it should not be claimed as original (at least can be shortened or omitted with references given). As another example, $\mathcal{J}_1$ is referred to as an “accurate” estimate, yet I am not sure if this is a standard notion; I agree that it should enjoy some good properties, such as unbiasedness (under the condition that the optimal $\phi$ exists), and perhaps consistency, etc, yet these properties require rigorous arguments to establish.
- The presentation of the content would benefit from additional polishing and thorough proofreading. The current draft contains a good number of typographical errors. Additionally, I find many of the citations and equation references in an inappropriate style (with versus without parentheses).

---

> ### Author Response · Authors · 2024-08-01
> **Author rebuttal**
>
> # Reviewer u9Aw
>
> Thank you for your insightful comments and engaging questions, which helped us improve the manuscript. We greatly appreciate the reviewers' acknowledgment that our work is novel, and our empirical evaluation thoroughly demonstrates the effectiveness of our algorithm.
>
>
> Here are answers to your concerns and questions.
>
> **Q. Quantification of errors when estimating $\mathcal{J}_1$?**
>
> >Accuracy of $\mathcal{J}_1$ as an estimator is frequently brought up (Section 3), yet I cannot find much concrete and precise support on it and, in fact, I am not sure what “accurate” means mathematically. To me, Theorem 2 suggests that $\mathcal{J}_1$ could be a reasonable estimate that is potentially unbiased (under conditions on $\phi$). To make such claims, I would expect some more rigorous argument that provides quantification of errors.
>
> >The top paragraph on Page 9 (above Section 4) seems vague and imprecise. For instance, the mode of convergence must be specified in the centered equation.
>
>
> We would like to thank the reviewer for raising this point. As suggested, we have tried to make a more precise argument to support our estimator $\mathcal{J}_1$. First, we added the mode of convergence as follows:
>
> > Given an optimal Kantorovich potential $\phi$, we use the law of large numbers to transform the original optimization problem into a learning problem, which is
> $$
> \frac{1}{n}\sum_{i=1}^n ( \phi(X_i) - \phi(Y_i) )\rightarrow E_{X\sim\mu}[\phi (X)] - E_{Y\sim\nu} [\phi(Y)]=W_1(\mu,\nu) \\ \quad\text{w.p. 1 with respect to $\mu\times\nu$,}
> $$
> as $n$ grows to infinity.
>
> In addition, we have added a paragraph, which quantifies the error using the Hoeffding inequality. We would refer to page 9 in the updated version.
>
> > The error can be quantified under suitable assumptions. For instance, as long as $\phi$ is bounded, the Hoeffding inequality yields that
> \begin{align*}
>     \mathrm{Pr} \left(\left|\frac{1}{n}\sum_{i=1}^n ( \phi(X_i) - \phi(Y_i)) - W_1(\mu,\nu) ) \right| \geq \epsilon \right) \leq 2\exp ( - Cn\epsilon^2 )
> \end{align*}
> for some constant $C>0$.
>
> Here, we assume the boundedness of $\phi$, which can be easily implemented in the algorithm. Theoretically, the boundedness of $\phi$ follows from the continuity of $\phi$ and the boundedness of the domain, which holds true in the space of images.
>
>
> **Q. The significance of Proposition 2?**
>
> We understand the reviewer's opinion on the importance of Proposition 2. Please allow us to elaborate on the significance of the proposition.
>
> Given $\mu \in \mathcal{P}(\Omega)$ and a set of empiricial data $\{X_1,...,X_m\}$ such that $X_i \sim \mu$, we seek for the best approximation $\nu$ of $\mu$ through opimization of $E[\mathcal{J}_1]$ which is approximated with empricial measure $\mu_m$. Thanks to the symmetry property of $\mathcal{J}_i$, one might choose $\mathcal{J}_i$ for $i=2,3$ as an objective function. Proposition 2 yields that replacement of $\mathcal{J}_1$ with $\mathcal{J}_2$ might lead to a poor approximation, which justifies the leveraging $\mathcal{J}_1$ as an objective function. The proposition claims that optimization of $E[\mathcal{J}_2]$ results in the unstable estimation of the target measure $\mu$ and such an observation is also supported by an empirical verfication given in Figure 2. We have added a more detailed explanation on pages 11 and 12 of the updated version.
>
>
> **> Proposition 1**
>
> We agree with the reviewer that Proposition 1 well-known. We have deleted the proof and referred to Villani (2008).
>
>
> **> Typos**
>
> We distinguish narrative citations and parenthetical citations, as suggested. In particular, equation references were corrected to ( ref{label}).
>
> > Citation styles: Narrative citations and parenthetical citations should be distinguished in their uses.....The same applies to equation references.
>
> All raised comments on typos have been addressed as suggested.

---

### Review · Reviewer_ZGYm · 2024-09-17

**Summary Of Contributions:**

The paper proposes to use c-transform methods to estimate the 1-Wasserstein distance. Furthermore, CoWGAN is proposed to enforce the 1-Lipschitz bound without the need for gradient penalty. Comparison with baselines shows faster convergence rate in estimating the Wasserstein distance and better enforcing the 1-Lipschitzness of the discriminators during GAN training.

**Audience:**

Yes

**Broader Impact Concerns:**

No additional concerns.

**Claims And Evidence:**

No

**Requested Changes:**

I would suggest the following adjustments:
1. Include other relevant baselines and report quantitative results, for example, IS and FID. Provide results on larger-scale experiments if possible.
2. Provide further explanation regarding the results in Figure 7. Specifically, the Lipschitz estimates for CoWGAN frequently exceed a value of 1, which raises concerns about how well the method enforces the 1-Lipschitz condition. And it is better to further clarify why the proposed method has better performance enforcing the 1-Lipschitzness of the discriminator, e.g.. through quantitive analysis.

**Strengths And Weaknesses:**

Strength:
1. The motivation of the paper is clear. The paper tries to address the limitations of WGAN-GP, which does not strictly enforce the 1-Lipschitz condition.

Weaknesses:
1. Firstly, the experiments are limited in size. Results are mainly conducted on small datasets with DCGAN. It is unclear if the proposed method works for larger-scale experiments.
2. Secondly, there is no quantitative comparison of generated images quality, for example, in terms of IS or FID.
3. Moreover, only WGAN-GP is included as baseline. Other relevant baselines, for example, SNGAN or Gradient Normalization [1], are not included for comparison.

[1] Wu, Yi-Lun, et al. "Gradient normalization for generative adversarial networks." Proceedings of the IEEE/CVF International Conference on Computer Vision. 2021.

---

> ### Author Response · Authors · 2024-10-12
> **Author rebuttal**
>
> # Reviewer ZGYm
> Thank you for taking the time to review our work. We appreciate your acknowledgment that the motivation for addressing the limitations of WGAN-GP is clear. We deeply value your feedback and are happy to address your questions. For all changes, please refer to the revised manuscript.
>
> **Q. Further explanation regarding Figure 7**
>
> > Provide further explanation regarding the results in Figure 7. Specifically, the Lipschitz estimates for CoWGAN frequently exceed a value of 1, which raises concerns about how well the method enforces the 1-Lipschitz condition. And it is better to further clarify why the proposed method has better performance enforcing the 1-Lipschitzness of the discriminator, e.g.. through quantitive analysis.
>
> Thank you for raising this important point. We agree with the reviewer that the Lipschitz estimates for CoWGAN occasionally exceed a value of 1. This behavior is likely due to the inherent nature of the Lipschitz constant and the fact that the supremum computation can be influenced by outliers, as it measures the largest difference in the function over the entire dataset. In other words, even a small number of outliers can inflate the estimate, leading to values greater than 1.
>
> In response to this concern, we have added histograms (see Figure 8 in the revised manuscript) showing the empirical density of the Lipschitz constant, calculated using 10,000 pairs of data from both real and synthetic datasets. These histograms provide a more detailed view of how the Lipschitz constant behaves in practice. As illustrated, for more than 98.5% of the data pairs, the Lipschitz constant remains less than or equal to 1, demonstrating that our method enforces the 1-Lipschitz condition well for the majority of the data.
>
> **Q. Comparison with other relevant baselines?**
>
> > Moreover, only WGAN-GP is included as baseline. Other relevant baselines, for example, SNGAN or Gradient Normalization [1], are not included for comparison.
>
> In response to the reviewer’s suggestion, we have included WGAN-GP with Spectral Normalization (SN) for comparison. As shown in Figure 7, the Lipschitz constants for WGAN-GP with SN are quite low, around 0.1 for the MNIST and Fashion-MNIST datasets. For CIFAR-10 and CelebA-HQ, the constants range between 0.2 and 0.5.
>
> Additionally, we have compared the adjusted discriminator loss (discriminator loss divided by the Lipschitz constant) with WGAN-GP with SN, as presented in Figure 9.
>
> We also conducted experiments with SNGAN (without the gradient penalty). However, since the generating performance of SNGAN was not comparable to the other methods, we chose not to include those results in the main comparison.
>
> > Include other relevant baselines and report quantitative results, for example, IS and FID.
>
> Thank you for pointing out this. In the revised manuscript, we include the Inception Score (IS) and the Fréchet inception distance (FID) with CoWGAN (ours) and WGAN-GP, WGAP-GP with SN, respectively, with CIFAR10 and CelebA-HQ datasets. Please see Table 2.

---

### Review · Reviewer_1Zwh · 2024-12-23

**Summary Of Contributions:**

The paper proposes a novel method for training Wasserstein Generative Adversarial Networks (WGANs) without using gradient penalties. The authors introduce two objective functions based on the c-transform and Kantorovich duality, which they claim can accurately estimate the Wasserstein distance between datasets. The method eliminates the need for gradient penalties and corresponding hyperparameter tuning, and it is demonstrated to be effective in generating competitive synthetic images using various datasets such as MNIST, Fashion-MNIST, CIFAR-10, and CelebA-HQ.

**Audience:**

Yes

**Broader Impact Concerns:**

~

**Claims And Evidence:**

Yes

**Requested Changes:**

- While the method avoids the computational complexity of directly computing the c-transform for large high-dimensional datasets, it still involves complex optimization steps that may be challenging to implement efficiently. Can the authors discuss this issue?
- There are many efficient methods for OT problems without adding penalty terms, such as IPOT. Can the author compare some relevant methods from a theoretical and experimental perspective, or explain the differences.
- While the theoretical foundation is strong, some parts of the mathematical proofs and derivations could benefit from clearer explanations, especially for readers who are not experts in optimal transport theory. Please modify.

**Strengths And Weaknesses:**

Strength:
- A novel method for training Wasserstein Generative Adversarial Networks (WGANs) without using gradient penalties based on optimal transport theory

Weakness:
- Lack of sufficient comparison with other no-penalty methods
- Need more clear organization of the proof process

---

> ### Author Response · Authors · 2024-12-27
> **Author's response**
>
> We deeply value your feedback and are happy to address your questions.
>
> **Q. Discussion of inherent difficulties of complex optimization**
>
> > While the method avoids the computational complexity of directly computing the c-transform for large high-dimensional datasets, it still involves complex optimization steps that may be challenging to implement efficiently. Can the authors discuss this issue?
>
> Thank you for raising this point. As you noted, our algorithm avoids the need to compute the $c$-transform over the entire dataset; instead, it only requires computation on minibatches. This greatly simplifies the algorithm and reduces its complexity.
>
> We also emphasize that our algorithm is designed without involving complex computational steps. However, we acknowledge the reviewer’s concern regarding the theoretical guarantees of convergence. While our experimental results provide empirical support, the paper does not comprehensively address convergence analysis, which needs to be further explored.
>
> **Q. Comparison with other methods not requiring gradient penalty terms**
> > There are many efficient methods for OT problems without adding penalty terms, such as IPOT. Can the author compare some relevant methods from a theoretical and experimental perspective, or explain the differences.
>
> We appreciate the suggestion to compare with gradient-penalty-free methods such as IPOT [1]. Our work shares a similar motivation in avoiding penalty terms but differs by leveraging Kantorovich duality through the $c$-transform. While IPOT focuses on accurate and stable estimation of the Wasserstein distance, our method directly enforces 1-Lipschitz continuity via a novel optimization process.
>
> To elucidate the difference, we have added the following in Section 1.1.1 (Estimation of Wasserstein distance), on page 2.
>
>
> > We also highlight the recent work by [1], which introduces the Iterative Proximal Optimal Transport (IPOT) algorithm. The algorithm efficiently approximates the Wasserstein distance without introducing a gradient penalty term. The authors further demonstrate that IPOT can also be applied to generative tasks. However, unlike our method, which directly enforces the 1-Lipschitz condition using Kantorovich duality and the $c$-transform, IPOT focuses on optimizing the transport plan without explicitly addressing the Lipschitz continuity of the discriminator.
>
> Another relevent baseline, Spectral Normalization (SN) [2] and its variants such as WGAN-SN have been added in the revised version as well. As shown in Figure 7, the Lipschitz constants for WGAN-GP with SN are quite low, around 0.1 for the MNIST and Fashion-MNIST datasets. For CIFAR-10 and CelebA-HQ, the constants range between 0.2 and 0.5. Additionally, we have compared the adjusted discriminator loss (discriminator loss divided by the Lipschitz constant) with WGAN-GP with SN, as presented in Figure 9.
>
> We also conducted experiments with SNGAN (without the gradient penalty). However, since the generative performance of SNGAN was not comparable to the other methods, we chose not to include those results in the main comparison.
>
> [1] Xie, Yujia, Xiangfeng Wang, Ruijia Wang, and Hongyuan Zha. "A fast proximal point method for computing exact Wasserstein distance." In Uncertainty in artificial intelligence, pp. 433-453. PMLR, 2020.
>
> [2] Miyato, Takeru, Toshiki Kataoka, Masanori Koyama, and Yuichi Yoshida. "Spectral normalization for generative adversarial networks." arXiv preprint arXiv:1802.05957 (2018).
>
> **Q. Presentation of a clearer explanation for the theory of optimal transport**
>
> > While the theoretical foundation is strong, some parts of the mathematical proofs and derivations could benefit from clearer explanations, especially for readers who are not experts in optimal transport theory. Please modify.
>
> Thank you for the input. In response to the reviewer's request, we have modified **Section 2.1. Optimal transport overview** to provide a detailed and intuitive introduction to attract broader audiences.
>
> The changes can be found in Section 2.1 on page 4 of the revised manuscript.
> > Under suitable assumptions on $\mu$ and $\nu$, we also have
> $$W_p(\mu,\nu) = ...$$
> ...It can be interpreted as the minimum cost of transporting the distribution $\mu$ to $\nu$ with respect to the metric given by $\|x-y\|^p$ for $p\geq 1$, which also can be used as the \textit{distance} between two distributions. However, solving this optimization problem is challenging in general.
> ...
> Finally, the Kantorovich-Rubenstein duality yields that
> $$
>      W_p^p(\mu, \nu)=...
> $$
> ..., we optimize over a scalar function $\phi(x)$, which is known to be efficient in high dimensions.
>
> We also have polished the proof of Theorem 3 and Proposition 2 for better presentation.
>
> The authors would appreciate any further comments and suggestions.

---

### Decision · Action_Editor_G4YK · 2025-02-04

**Recommendation:** Reject

**Comment:**

The paper proposes CoWGAN, a method for training Wasserstein Generative Adversarial Networks (WGANs) without using gradient penalties. By leveraging the c-transform and Kantorovich duality, the method introduces new objective functions that accurately estimate the Wasserstein distance while enforcing the 1-Lipschitz constraint without requiring gradient penalties or hyperparameter tuning. Experimental results on datasets such as MNIST, Fashion-MNIST, CIFAR-10, and CelebA-HQ demonstrate improved Wasserstein distance estimation, faster convergence, and high-quality image generation.
    The authors claimed that the method eliminates the need for gradient penalties and corresponding hyperparameter tuning. However, as a reviewer pointed out, the experiments are limited to small datasets and small network architectures. It would be convincing if the authors have compared with StyleGAN-series or BigGAN, and consider datasets like FFHQ. Some recent GAN papers have been using ImageNet-256, but for this paper, ImageNet-256 is too much effort and not needed. Please check recent GAN papers for the mainstream test setting.
     Oeverall, I thnk the claim that "eliminating the need for gradient penalties and hyperparameter tuning" is not justified by existing experiments. Thus I recommend rejection.
  Nevertheless, the authors may resubmit with a substantial amount of additional experimentation that can help justify their claim.

**Audience:**

Yes. The work proposed a modification to GAN. It is likely of interest to readers working on the theory of generative models.

**Claims And Evidence:**

The authors  claimed that "the method eliminates the need for gradient penalties and corresponding hyperparameter tuning". However, the experiments are limited to small datasets and small network architectures. Note that in training GANs, the larger model and dataset size, the harder to train and thus the stronger  need for extra techniques like gradient penalties. Thus, the claim of "eliminating the need for gradient penalties and hyperparameter tuning" is not justified by existing experiments.

**Resubmission Of Major Revision:**

The authors may consider submitting a major revision at a later time.